# An Overview of the Policy and Market Landscape of Tobacco Production and Control in Mozambique

**DOI:** 10.3390/ijerph18010343

**Published:** 2021-01-05

**Authors:** Nicole Nguenha, Benedito Cunguara, Stella Bialous, Jeffrey Drope, Raphael Lencucha

**Affiliations:** 1Independent Researcher, Av. Vladmir Lenine #2081, Flat 1.4, Maputo P.O. Box 55, Mozambique; cunguara@gmail.com; 2Social and Behavioral Sciences Department, School of Nursing, UCSF, San Francisco, CA 94143, USA; Stella.Bialous@ucsf.edu; 3Division of Health Policy and Administration, School of Public Health, University of Illinois at Chicago, 1747 West Roosevelt Rd., Chicago, IL 60607, USA; jdrope@uic.edu; 4School of Physical and Occupational Therapy, Faculty of Medicine, McGill University, 3630 Promenade Sir William Osler, Montreal, QC H3G 1Y5, Canada; raphael.lencucha@mcgill.ca

**Keywords:** tobacco control, tobacco farming, public policy, agriculture, agricultural production, governance

## Abstract

*Background:* Tobacco growing has been considered a mainstay of Mozambique’s economy, but there is a dearth of analysis of the tobacco policy landscape in the country. *Methods:* Review of government and non-government documents and academic papers addressing Mozambique’s tobacco-growing history, the changes in the political economy of tobacco, and health policies addressing tobacco use and prevention of noncommunicable diseases. *Results:* Despite its tobacco growing and exporting history, the contribution of tobacco to the economy has been in steady decline in the past two decades, including in the areas dedicated to growing. At the same time there has been an increase in multinational control of the tobacco economy. In parallel, Mozambique’s commitment to addressing the growing burden of noncommunicable disease and accession to the Framework Convention on Tobacco Control indicate a potential for internal government tensions to balance immediate economic interests with long term health goals. *Conclusions:* With the decline in tobacco share of the overall economy, Mozambique may be well-positioned to explore alternative, sustainable livelihoods for farmers that grow tobacco, but it must overcome inter-sectoral barriers and advocate for a whole of government approach to address the health and economic impact of tobacco.

## 1. Introduction

Tobacco control requires action at different levels and across different sectors. The slate of policies required to reduce demand are well known and supported by decades of evidence of effectiveness [1]. Taxation measures, smoke-free environments, advertising bans, graphic warning labels, and age restrictions have all, when implemented as part of a comprehensive approach to tobacco control, been proven to reduce consumption at the population level. This policy slate is also enshrined in the Framework Convention on Tobacco Control (FCTC), an international legal agreement with 182 parties committed to implementing its provisions. The challenges of implementing comprehensive tobacco control lie in part within governments. Governments often have conflicting mandates and aims across sectors, where the health sector has a singular focus on implementing the most comprehensive tobacco control measures to reduce consumption, whereas the economic sector must consider the economic livelihoods of tobacco farmers and the macroeconomic dynamics of revenue generation and foreign exchange earnings [2,3,4,5]. This perceived tension between economic and health aims, narrowly conceived, has created barriers to tobacco control in many countries, especially where tobacco is grown, processed or manufactured. Many tobacco-producing countries like Mozambique have extensive histories of tobacco growing that pre-date the rise of the global tobacco control movement [6,7]. These histories come with institutional legacies in which tobacco production is embedded in and supported by the government [8]. In order to move towards stronger tobacco control measures, and the implementation of the demand and supply measures in the FCTC specifically, it is critical to understand the policy landscape across sectors. The principal rationale for pursuing this understanding is that comprehensive tobacco control cuts across sectors and require the participation and support of not just health ministries but also finance, industry, agriculture, trade and others [9]. To understand this policy landscape is to better understand the points of intervention and potential existing conflicts and complementarities across government policies. This research aims to chart the policy landscape in Mozambique, which ranks as the fourth-largest tobacco producer on the African continent and a Party to the FCTC since 2017. This paper draws from a comprehensive document review process supplemented by analysis of existing economic data such as tobacco production and export. Research on the context of tobacco (control and production) policy in Mozambique, particularly from a historical and political economy perspective, is still nascent. In addition to charting the policy landscape, this paper explores policies and programs promoting alternative cash crops for farmer livelihoods, as well as potential tensions that tobacco supply may have on tobacco control.

## 2. Materials and Methods

Figure 1 outlines the methodology. Data collection was mainly dependent on secondary data sources, along with some informal discussions with stakeholders where necessary and possible. A previously prepared analytical framework for the policy and document analysis [10] focused on (i) incentives for tobacco production/priority crops as highlighted by the government, (ii) historical evolution of tobacco production, (iii) private sector involvement and (iv) public health involvement (tobacco control). A desk review of documents from the year 2000 onwards was guided by this framework and involved collecting, organizing and synthesizing the available information related to the political economy of tobacco in Mozambique. The desk review aimed to better understand the country context from both the general crop and tobacco production perspective, which is generally under Mozambique’s Ministry of Agriculture and Rural Development (MADER, in Portuguese), as well as from the tobacco control perspective, which is overseen by the Mental Health department in the Ministry of Health (MISAU, in Portuguese). Government officers from both MISAU and MADER were consulted to ensure that all policies, regulations and/or strategic documents were included in the analysis. Within this process, an overview of all agriculture-related and health policy documents was completed, and key components of these documents as it related to the research questions were summarized in a matrix. Additional relevant documents analyzed for the purpose of completing the desk review included: peer-reviewed articles, public data surrounding agricultural production and exports from the National Statistics Bureau (INE, in Portuguese) and various nationally representative agricultural surveys, laws and regulations, news/media outlets and research reports. A stock-taking report of agriculture-related papers was done in 2011, and one of the outputs was a list of all relevant agricultural studies on Mozambique [11]. The list of tobacco-related literature on Mozambique was used to supplement the current review.

## 3. Analysis

### 3.1. Historical Agribusiness Trajectories

Tobacco was first introduced in Mozambique in colonial times by Portuguese traders and was traditionally grown by African peasants for domestic consumption. At the beginning of the 20th-century, tobacco became a commodity in Mozambique that could be exchanged for clothing and other consumer goods on the market as well as a source of income to pay mussoco, the colonial tax. The districts of Malema and Ribàué in the province of Nampula, today is known as two of the most agriculturally productive districts in the country, became the center of tobacco production in settlers’ farms, a production that reached an average of approximately 3000 tons per annum at that time. Without agricultural assistance or access to credit, the relative success of tobacco production was largely dependent on the mobilization of forced labor for Portuguese farms. Navohola [12] calculated that in 1948 most of the workers came to work on the tobacco plantations through chibalo (forced labor during the colonial regime) in Ribàué and Malema. However, the volumes produced were limited, and Mozambique was a net tobacco importer, mainly from Angola and the USA [7]. Between 1941 and 1960, tobacco production increased, and the land-used to grow tobacco expanded. Those charged with the colonial agricultural strategy began to take an interest in ensuring that local production grew to replace imports and to supply the Portuguese metropolis.

Prior to obtaining independence from the Portuguese regime in 1975, tobacco farms flourished in central Mozambique due to the inexpensive forced labor and expropriated land [13]. Tobacco cultivation continued, mainly in the nationalized state farms of Manica and Nampula. In the following decade after independence, however, state tobacco farms were malfunctioning due to the impact of the civil war in 1977, which aggravated problems of poor coordination and mobilization. Aspects such as prices, financing and conditions for recruiting labor were managed by central authorities in many cases, not effectively managing problems that arose. As a result, the state initiated the selling of state farms in 1985, selling over 40,000 acres of land to the private sector by 1993 [7]. In the period following the end of the civil war, tobacco cultivation expanded in Mozambique in terms of production volume and was also introduced in provinces and districts where it had not previously been cultivated. In contrast to the colonial period, and in which commercial tobacco production had been restricted to state farms, after independence, tobacco was adopted mainly by Mozambican farmers using family labor and private land [7]. After the civil war that took place immediately after independence and following the process of divestment of state farms and the reform of seed supply organizations, producers who needed credit and assistance for the production of cash crops had to stop production due to the lack of capital and underdevelopment of local private credit unions. In order to deal with the existing bottleneck caused by the lack of financing and access to markets, a variation of the old system of concessions was adopted, this time in the form of interconnected input and production markets, in which cash crop traders provided credit to producers. In some regions of Mozambique, and for cultures such as tobacco and cotton, the only channel of access to credit assistance was the private sector [7]. The largest distribution of private credit and extension services in Mozambique in the past few years has been for tobacco producers. According to a recent survey from the Ministry of Agriculture and Rural Development, in 2017, only 2 percent of non-tobacco-producing farmer households had access to credit compared to 21.7 percent of tobacco-producing farmer households [14].

### 3.2. Current Country Context

The Republic of Mozambique is located in South-East Africa and is utilized by neighboring landlocked countries to access external markets. Mozambique has a population of 28.9 million, of which 84.9% are active in the formal economy. Smallholder farmers (SHFs) account for 86.2% of the rural population. Mozambique’s arable land accounts for about 36 million acres, and agriculture accounts for up to 80% of the nation’s economic workforce, though it only accounts for 25% of the nation’s GDP [15]. In this context, accounting for over a quarter of the total export value of agricultural commodities in Mozambique, tobacco is considered to be one of the country’s most important cash crops. The tobacco industry in Mozambique contributes to more than 1.9 billion *Meticais* annually in tax revenue to the state [16].

With roughly 98% of tobacco being produced by SHFs [9], the crop is grown in a concession system with Mozambique Leaf Tobacco (MLT), a local subsidiary of the multinational Universal Corp [17]. The company operates in four provinces in central and northern Mozambique: Zambézia, Manica, Niassa, and Tete (See Figure 2 for production by province).

Between 72,000 and 124,000 households grow tobacco out of 3.9 million farming households total nationwide, with higher concentrations of tobacco farmers in the Tete and Niassa provinces, which together in 2017 accounted for approximately 78% of the total number of growers. The total number of farmers varies considerably annually, however, and data from national agricultural surveys conducted by MADER show a consistent decline in Zambézia where alternative livelihoods have been promoted. The export data presented in Figure 3 illustrates that, as of 2015, Mozambique had a diverse international market for raw tobacco, with Belgium, Turkey, Poland, Netherlands and Ukraine having the highest tobacco export values.

According to data from the Ministry of Agriculture [19], which also measured data for other cash crops, such as cotton, sugarcane, sunflower, sesame and soy, most medium to small holdings that produced tobacco sold 100 percent of their crop, with the exception of Cabo Delgado and Tete, reflecting the crop’s high demand, and possibly, an efficient supply chain (Figure 4).

### 3.3. Production Incentives for Smallholder Farmers

In addition to job creation opportunities and high demand from a diverse market, the incentives of tobacco production, specifically for rural smallholder household welfare and farming systems, have been plentiful. Data by the national agricultural sample survey (TIA) in 2002 revealed that at the time, the incomes of tobacco growers were much higher than those who did not produce tobacco [20]. In the same study, controlling for a set of characteristics including the gender of household head, age, and education, researchers compared the total household income of tobacco smallholders to those who do not grow tobacco and did the same for those who grow cotton as a cash crop. The results demonstrated that this period saw a rise in income of SHFs who grew tobacco, about 29% higher than that of non-growers, while the income of cotton growers was only 5% higher than those that did not farm it. Given the high involvement of family labor in tobacco growing, this figure likely does not reflect the true profit of tobacco growing in relation to other crops, given that the analysis did not account for household labor. The findings also indicated that tobacco SHFs were less likely to be poor and to show improvement in their well-being over the past 3 years compared to households that did not grow tobacco. Preliminary empirical data suggests that the additional income from sales, as well as the additional fertilizer residuals left in the soil from tobacco production, can improve food security [21,22]. The additional income garnered from tobacco sales as well as effective agricultural practices, such as crop rotation, has increased the productivity of maize after tobacco production. In general terms, tobacco producers tend to benefit from better access to extension services, including inputs, pesticides and fertilizer, which tends to contribute to the positive yields, including that of non-tobacco crops. The combined direct and indirect effects of tobacco production are stated as one of the reasons why Tete province managed to achieve the fastest rate of increase in rural household incomes in Mozambique in the few years following 1996 [21]. Access to such services for tobacco producers were of national interest and in alignment with the Strategic Plan for Agrarian Sector Development 2011–2019 (PEDSA, in Portuguese) [23]. To support the implementation of the PEDSA, there are additional strategic documents that operationalize the PEDSA, including the Agricultural Livestock Marketing Integral Plan (PICA, in Portuguese) [24] and Agricultural Sector National Investment Plan (PNISA, in Portuguese) [25]. We have included a Appendix A highlighting the political framework listing relevant agricultural policies/programs (2000–onwards), including policies that promote tobacco.

### 3.4. Regulation and the Tobacco Market

Tobacco production in Mozambique is regulated by the Tobacco Promotion, Production and Marketing Regulation (Ministerial Diploma 176/2001) [26] and by the contracts established between the national government and the tobacco concession companies. This concessionary system was implemented after the divestment of state farms. The Tobacco Regulation sets out the principles that govern concessions and establishes the role of the various stakeholders involved. The Mozambican model is composed of three interrelated elements: (1) the interconnected input and production markets, (2) a production scheme under contract as the predominant form of production, and (3) the adoption of monopsonistic territorial concessions.

The first of these elements, the interconnection between the input and production markets, served as a solution to the lack of credible sources for agricultural producers [27,28]. The background to this situation was that traditional financial institutions were unable to offer commercial credit to producers because the small scale of the credit that farmers asked for increased overall transaction costs; the lack of markets prevented the use of land as collateral generally, and debts were difficult to collect. One way around this problem was for agricultural traders to directly offer producers credits for production and deduct payments when buying the harvest. Unlike banks, agricultural traders have a direct link with producers, and traders would thus benefit from improved quality and production generated by access to agricultural credit. Therefore, the interconnected markets allow the use of the future harvest as a guarantee for the repayment of production credit [28]. To this end, traders sign production contracts with each of the producers, in which they advance in-kind credit in the form of inputs for production (seeds, fertilizer, pesticides and other production materials) and commit to buying the final product, even though often not all the tobacco is purchased leaving farmers with a surplus. At the time of the purchase, the merchant deducts the initial credit amount from the payment the farmer receives for tobacco. Competition between companies creates an incentive for traders to offer attractive prices to producers to whom they have not provided credits, taking advantage if so, free of charge, of the lenders’ investment. With alternative buyers, the interconnected markets have higher rates of strategic default, as has been documented in the case of the cotton sectors in Ghana and Mozambique [28,29]. To avoid the negative effects of strategic non-compliance, some county governments have opted for the creation of territorial concessions, in which traders receive concessions in different regions, eliminating competition between operators [7].

In 2005, there were about 129,000 agricultural producers with tobacco production contracts. Of these, there were 34,813 producers in Niassa and 43,464 in Tete. MLT was the main buyer and had concessions in Tete and Manica. According to the 2001 Tobacco Regulation, under article 2, there are four classes of economic operators in the tobacco production space (see Figure 5 for classification details): Class I (family sector), Class II (non-self-employed farmers), Class III (self-employed farmers) and Class IV (promoters and traders). Producers engaged in agriculture under contract are classified in “Class I” if they do not use salaried labor and in “Class II” if they do. Independent farmers who are not bound by production contracts and are free to negotiate the price for their products are part of “Class III,” and leaf tobacco developers and traders who provide credit and operate the regional concessions comprise “Class IV”. Tobacco trading companies of “Class IV” apply annually to renew their concessions to the Provincial Directorates of Agriculture. Operators must submit a production proposal and specify inputs, investment and reforestation plans, as well as a final execution report after the harvest and the marketing season. Contract signatories are strictly prohibited from selling their tobacco to third parties not included in the contract. Tobacco sold outside the contract can be confiscated and handed over to its legitimate dealer in accordance with Ministerial Diploma 176/2001. Tobacco is sold by farmers at different price levels, with “premium price” being the highest. In Mozambique, the price set by the largest tobacco concessionaire, MLT, for the purchase of tobacco from SHFs is automatically never sold at a premium price and therefore is, by default, set to sell at lower prices [30].

In all tobacco producing districts, it is stipulated that an arbitration committee must be formed, with representatives of all producers, buyers, district government and community, to resolve differences between farmers and companies regarding the sorting, classification and marketing of tobacco. The MADER has an inspection body to ensure compliance with the tobacco regulation by all parties. By the mid-1990s, there were 8 main companies in the tobacco trading space in Mozambique, including DIMON, STANCOM, MLT, SONIL, JFS, Mosagrius, as well as two joint venture companies, Shancom and Stancom. Three large multinational tobacco trading companies, Dimon Inc., Stancom and MLT (the local subsidiary of Universal Corp.), started operating in Mozambique in the mid-1990s, joining companies such as JFS and SONIL, which have worked in Mozambique since the colonial period but were new to the tobacco trade. Stancom was present in Manica, and as a subcontractor for SONIL and Mosagrius, in both Niassa and Cabo Delgado province. Dimon worked in Tete, Manica and Sofala. The other major player, JFS, was a Portuguese Mozambican business group with tobacco and cotton contracts in Nampula, Niassa and Cabo Delgado province [7]. Although the Mozambican model started with several operators as described above, in 2006, it was reduced to a de facto national monopoly, in which a single company, Universal’s subsidiary MLT, dominates the purchase and processing of tobacco leaf, although marginal traders remain.

The mid-2000s marked a turning point in the trajectory of the tobacco market in Mozambique. There were three developments that determined, from then on, the evolution of the sector: technological developments such as the creation of a processing factory in Tete province, which ended the need to send raw tobacco to Malawi for processing; the withdrawal of the Chifunde concession, which led to the departure of Dimon Inc. from Mozambique; and the failure of tobacco production in the province of Manica. In the early 2000s, tobacco from Mozambique had a consolidated presence in the international market, but it had to be exported to Malawi or Zimbabwe, to be processed, and reexported through the port of Beira city in Sofala province since there were no processing facilities in Mozambique. This resulted in additional transport costs and loss of revenue. An internal proposal was studied at the MADER to introduce an export tax of 20% on the value of raw tobacco to force commercial companies to invest in a leaf-cutting infrastructure in Mozambique [21].

An internal document from 2004, prepared by the National Directorate of Agrarian Services of MADER was quoted by Benfica and colleagues [21] and argued that this tax would lead buying companies to invest in processing facilities and thus create job opportunities and new sources of tax revenue from the income tax paid for additional processing labor. The export tax proposal, however, had a more complex context. In February 2003, MLT, the largest purchasing concessionaire, had started the construction of a 50 million USD processing unit in Tete province, with a capacity to process 50 thousand tons per year [21]. Considering that MLT was, at the time, the biggest buyer and that total production in 2003 was 37,051 tones, it is likely that MLT wanted to ensure that the installed capacity would not be underutilized. Other tobacco companies did not buy tobacco on a scale that justified the installation of processing infrastructure. The introduction of an export tax in this context would have forced all tobacco producers to process tobacco at the unit built by MLT. The MLT processing unit was opened in 2006, and the export tax policy ended, but in 2005, the government announced that the tobacco concession in Chifunde, in Tete province, would be transferred from Dimon, at that time already merged with Alliance One, to MLT. Chifunde was the largest concession controlled by Alliance One, and its loss jeopardized the viability of its work in Mozambique. In May 2006, Alliance One announced that it would abandon all its concessions as of the 2007 agricultural season and began to close activities invoking political interference. At the time, 500 direct workers lost their jobs, although many ended up being reabsorbed by MLT. The decision to transfer the concession to MLT was initially seen as a reward for the willingness to invest in processing [31,32]. Only in 2010 did it go public that the MLT was behind the proposed export tax and the transfer of the Chifunde concession. In the following years, Mozambique stopped exporting raw tobacco material to neighboring countries for processing and became a country that exported processed tobacco ready to send for cigarette manufacturing. Additionally, Mozambique no longer has the initial eight registered companies, as only two, MLT and SONIL, remain in-country [7].

In 2019, a media source published that China was due to import approximately 60,000 tons of tobacco from Mozambique until February 2020, making them the second-largest tobacco importer. Mozambique’s current President, Filipe Nyusi, stated that the sale of these products would improve the livelihood of local farmers [33]. To confirm where this is, in fact, true requires further research. To date, there have not been any publications analyzing the economic livelihoods of tobacco farmers after this new market for tobacco leaf grown in Mozambique. Findings from Zimbabwe suggest that the China National Tobacco Company has renewed and facilitated the expansion of tobacco growing there [34,35]. It is likely that CNTC could have a similar impact in Mozambique.

### 3.5. Alternative Cash Crops for Farmer Livelihoods

As stated in earlier sections, the general consensus for producing tobacco is to improve farmer livelihoods. Although cotton and tobacco have a long history as cash crops in Mozambique, there are other cash crops that have recently been introduced to farmers. Soybeans are one example of a crop grown as an alternative to tobacco. The percentage of growers of tobacco at the national level declined slightly (Table 1) as farmers embraced new alternatives. TechnoServe, funded by the Embassy of the Kingdom of the Netherlands (EKN), implemented the Seed Multiplication project to Empower Small Commercial Farmers (SM4ESCF) from 1 March 2016 to 31 January 2019. The overall objective of the project was to increase the productivity and profitability of SHFs and small commercial farmers (SCFs) in Zambézia Province, resulting in financial benefits for these rural farming communities. Specifically, the project sought to build a strong and sustainable local seed and service provider network in Upper Zambézia.

The project invested heavily in extension services, machinery, post-harvesting technologies, and a seed factory. In 2002 about 3.5% of farmers in Zambezia cultivated tobacco, and no one grew soybeans. By 2017 the percentage of tobacco growers in Zambezia had declined to 0.8%, and there was a shift to soybean production (0% in 2002; 4.6% in 2017) attributed to the guaranteed market for soybean seeds, which, unlike tobacco, sells at a premium price at the factory, and due to demand for soybean that is sold to the poultry industry both locally and in neighboring Malawi. As previously stated, the tobacco contribution to farmer livelihoods in Mozambique is unknown. From both a public health and economic perspective, the cash crops outlined in Table 1 could be explored as healthier alternatives to tobacco, especially since there is a guaranteed market for some of these crops. Additionally, in accordance with the agricultural operational marketing plans (2018) developed by the Government for Zambézia province, the production of these alternative crops, such as sunflower, sesame and cotton, is, indeed, of government interest.

### 3.6. Tobacco Control (Ministry of Health)

The Ministry of Health, under the mental health department, has outlined tobacco control strategies in both national decrees and strategic documents. Under the decree for Consumption and Commercialization of Tobacco (Decree 11/2007), there are various limitations relating to the commercialization, consumption, as well as demand and supply reduction for tobacco, including the banning of partnerships/cooperation between tobacco companies and public health campaigns of any kind (Article 13). Relevant articles limiting the supply and sales of tobacco under Decree 11/2007 include Article 10, which explicitly prohibits the sale and publicity of tobacco in various establishments, including, but not limited to, health, education and public administration institutions; and Article 12, which prohibits the sale of tobacco products to minors (anyone under the age of 18). Relevant articles to decrease tobacco demand include the following: Article 3 provides for a modest increase in tobacco taxes; Articles 7 and 8, which outlaw general tobacco advertising as well as false marketing that outline false benefits related to tobacco; and lastly, Article 9, which highlights demand reduction measures related to tobacco dependence and cessation as the focus.

A common government interest that has been consistent in the decree and later translated in relevant national strategic documents concerning tobacco is the importance of awareness-raising of the negative impacts of tobacco use to the general public. Key policy documents including the Strategic Plan for the Health Sector (2014–2019) [36], National Strategic Plan for the Prevention and Control of Non-Communicable Disease (2008–2014) [37], National Plan for Cancer Control (2019–2029) [38] and the Mental Health Action Plan and Strategy (2007–2015) [39] have awareness-raising activities in relation to tobacco use included in their narratives and/or strategic objectives. In accordance with the indicators highlighted in the policy documents, the expected impact is to increase awareness surrounding the risks and negative impacts of tobacco.

The concessionaire that dominates the growing and processing of tobacco in Mozambique, MLT, has various social corporate responsibility programs. However, most are related to discouraging child labor practices, enhancing good agricultural labor practices, financial literacy, and increasing access to clean water. The program most related to tobacco control policy in Mozambique is an HIV program [17], due to the fact that it also focuses on tuberculosis as tobacco smoking often causes health problems in tuberculosis patients. However, it is unclear whether the messaging is around tobacco control, and while tuberculosis awareness seems to be a component of the program, it is not its central focus.

According to the government officials responsible for leading the multi-sectoral tobacco control working group under the Ministry of Health, and the mandate of the National Inspection of Economic Activities (INAE, in Portuguese), INAE ensures that all economic activities are completed in compliance with the law in order to create a good business environment in the country. INAE has been granted authority to work together with the Ministry of Health, through inspections, in order to ensure that the in-country tobacco control regulations are followed. It is also relevant to highlight that there is not much information about economic sectors taking up tobacco control from the health perspective, nor are there regular inspections by the Health Ministry or in collaboration with INAE to ensure consistent implementation of tobacco control regulations in economic sectors. Therefore, further stakeholder consultations with the private sector are advised to gather more data. In-depth interviews with decision-makers are also needed to explore this dynamic.

Under the auspices of the Ministry of Health, the aforementioned multi-sectorial working group for tobacco control was created to regulate the consumption and commercialization of tobacco in Mozambique. The working group which was mostly responsible for the creation of Decree 11/2007, which aimed to follow the FCTC framework, consists of focal points from all government ministries, civil society, UN (WHO, UNDP) and customs, was formed, and later became functional in 2005 after Mozambique signed the FCTC in 2003. Although Mozambique was indeed a signatory to the convention, it would be years to come until the government officially ratified the FCTC. Being one of the few tobacco-producing countries that had an existing tobacco control instrument prior to the ratification of the convention, the multi-sectoral working group continued to work with decree 11/2007 as a base for tobacco control advocacy, which led to the eventual accession to the FCTC on 2 November 2017. Granted that the ratification of the FCTC was a success, a high-level government official we spoke with emphasized that although the tobacco leaf MLT purchases are intended for exportation, it still returns Mozambique as manufactured cigarettes making them readily available for consumption [30]. It is also important to note that all neighboring countries have much higher cigarette prices and taxes than Mozambique. Due to this fact, cigarette smuggling out of Mozambique is a common occurrence [40]. Given the shift of general consumption of tobacco from high-income countries to low and middle-income countries (LMICs); as well as the projection that by 2030 more than 80 percent of the disease burden from tobacco will fall on LMICs [10], it is important to keep in mind that the supply of tobacco that stems from production may pose potential competing pressures on health sectors goals as it relates to tobacco control not only in Mozambique but in other LMICs.

## 4. Conclusions

This paper outlines the tobacco policy landscape across time with an added emphasis on the private enterprise operating to facilitate tobacco production. Not surprisingly, we find two parallel and conflicting approaches to tobacco. On one hand, the government continues to approach tobacco as an important economic commodity. This approach stems in part from a historical legacy of tobacco production that has increased since the 1950s onwards. However, what we find is that tobacco is not a dominant cash crop and is mainly situated in two regions. The findings suggest that shifts are currently taking place to scale-up other cash crops like soybean, which has demonstrated a slow, but consistent increase in production. What we also see in this analysis is a theme that emerges consistently from other tobacco-growing countries that are considering alternatives. The supply chain of tobacco tends to be robust and incentivizes production by establishing contractual relationships for the provision of inputs, transportation to market and a guaranteed market for leaf sales [41]. We find that in Mozambique, like neighboring countries such as Zambia, Kenya and Malawi [42], the tobacco supply chain is well developed and has key mechanisms, such as credit and inputs, which keep farmers growing this commodity. The existence of an arbitration committee in the tobacco-growing regions can help remedy conflict, particularly as tobacco companies are notorious for undercutting tobacco farmers with unfair grading schemes and low leaf prices while bolstering their credibility in the eyes of the public through corporate social responsibility activities [43,44]. One challenge is the concentrated power within MLT and the embedded and seemingly positive relationship with government agencies.

In the end, the tobacco control community approaches tobacco supply from a unique perspective. This perspective starts from the premise that tobacco should be phased out both as a consumer product and as an agricultural commodity. This perspective is not often one that is taken in the economic and agribusiness sectors [5,9,45]. The entry of China National Tobacco Company in Mozambique illustrates that production will likely continue to be guided by economic rather than health protection considerations. To a large extent, the market will continue to shape production, with new export markets affecting the scale of production. However, the economic situation for tobacco is projected to be precarious in terms of both demand and future prices, leading major tobacco growing countries like Malawi to actively pursue alternatives as a sustainable economic strategy [5]. Despite the historical presence of tobacco in Mozambique, our findings suggest that there may be multiple paths to alternatives. The economic viability of other cash crops and the programs that have demonstrated efficacy in supporting the supply chain for crops like soy appear promising. Despite the relatively marginal economic contribution of tobacco to the national economy, tobacco is highly concentrated regionally. A regional focus, both in terms of policy and alternatives, will be vital for tobacco control efforts. It is noteworthy that the tobacco control focal point person would suggest that the government’s support for tobacco production is a major barrier to tobacco control in the country. The points of divergence can serve as points of intervention and further analysis to better understand how a mutually acceptable strategy of tobacco reduction (demand and supply) can be pursued in a way that does not jeopardize the economic livelihoods of farmers and the economic contribution to the broader economy of the country while recognizing the health consequences of consumption is a slow-moving disaster to the lives of consumers and to the wider health system.

## Figures and Tables

**Figure 1 ijerph-18-00343-f001:**
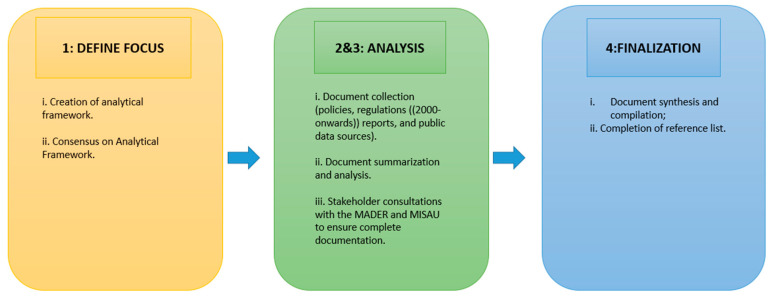
Methodology outline for desk review.

**Figure 2 ijerph-18-00343-f002:**
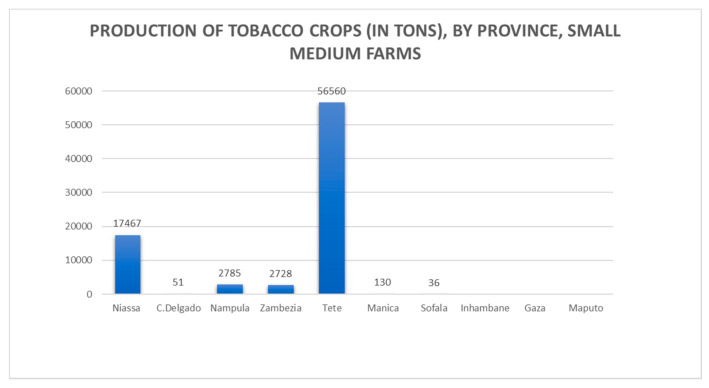
Production of tobacco crops (in tons) by province.

**Figure 3 ijerph-18-00343-f003:**
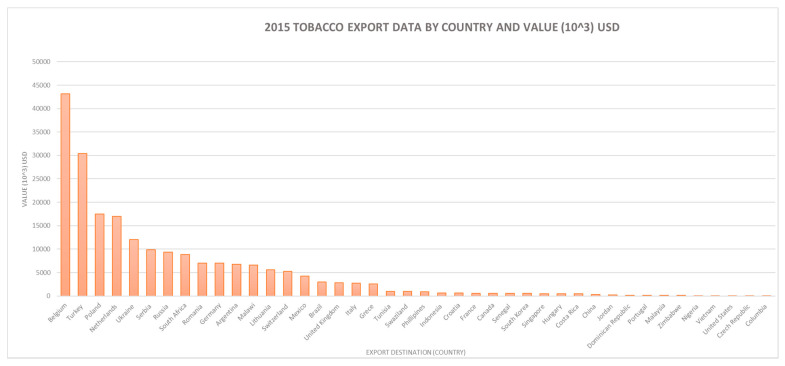
Key export markets by country (Instituto nacional estatistico, 2015–2016) [18].

**Figure 4 ijerph-18-00343-f004:**
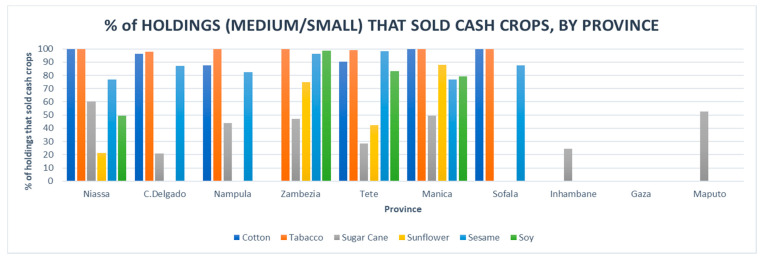
Percentage of holdings that sold cash crops by province (Anuario Estatistico, Ministerio de Agricultura e Segurança Alimentar, 2015) [19].

**Figure 5 ijerph-18-00343-f005:**
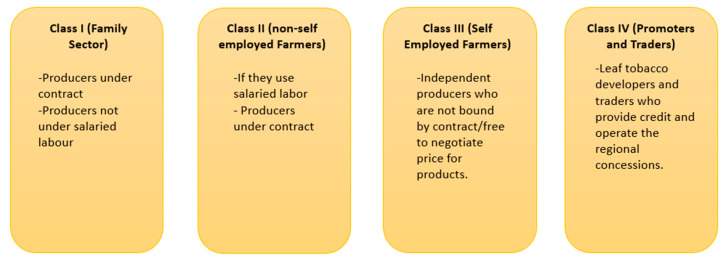
Four classes of economic operators in Mozambique’s tobacco production space.

**Table 1 ijerph-18-00343-t001:** Percentage of smallholder farmers who cultivated cash crops by year in Zambezia province, Mozambique (source: MADER survey data) [15,18].

Crops	Percentage of Smallholder Farmers Who Cultivated Crops (by Year)									
	2002	2003	2005	2006	2007	2008	2012	2014	2015	2017
Cotton	7.2	5.2	6.6	6.1	5.1	3.9	6.3	4.2	3.0	2.4
Tobacco	3.8	3.1	3.1	3.7	2.6	2.6	1.5	2.0	2.2	2.0
Sisal	0.2	0.0	0.0	0.1	0.1	0.1	0.0	0.0	0.0	0.1
Tea leaves	0.1	0.0	0.0	0.0	0.0	0.1	0.0	0.0	0.0	0.0
Sugar cane		13.8	8.1	8.6	9.1	6.1	4.4	3.8	3.0	2.9
Sunflower	2.5	1.2	0.5	1.4	1.4	0.9	0.6	0.6	0.7	0.8
Sesame	7.7	5.2	8.1	7.2	6.7	7.4	8.2	10.2	10.0	7.2
Soybeans	0.3	0.4	0.7	0.8	0.9	0.8	0.9	1.2	2.1	2.4
Paprika	0.2	0.1	0.1	0.1	0.0	0.0	0.0	0.0	0.0	0.0
Ginger	0.4	0.0	0.1	0.0	0.0	0.0	0.0	0.0	0.0	0.0

## Data Availability

All data has been published as Appendix A and was collected from public sources.

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
