# Peer review of "An Overview of the Policy and Market Landscape of Tobacco Production and Control in Mozambique"

_ijerph, 2021, doi:10.3390/ijerph18010343_

Round 1

Reviewer 1 Report

This study gives an overview of the production of tobacco and tobacco control development in Mozambique. It effectively combines history, policy, and interviews with key people to fill in the necessary details. I only have a few comments to increase readability and point out potential editing snafus.

Overall, I don't think that the footnotes serve to make the article easier to understand or read, and the text in them can be either omitted or moved: There is no footnote 1 in the current version, 2 could be moved up to the text of the document, 3 could be an endnote as it is a citation, 5 appears in the text but doesn’t have a footnote at the bottom of the page, 6 could be in the text of the document, 7 could be in the text of the document, 8 could be in the text of the document, and 9 can be deleted because it appears that with current formatting nothing is highlighted in yellow.

The second reference to the National Statistics Bureau on line 98 is not capitalized and presents the same shorthand term for the Bureau from line 82. Consider either capitalizing its name throughout or using the shorthand term. Similarly, you gave a shorthand acronym for the Ministry of Agriculture and Rural Development on line 74 but then generally used the full name of the Ministry throughout -- it seems that it might be more efficient to use the shorthand term.

Table 2 appears to be on the wrong line of the page, it is too far down the page and should be after line 314 and before 315.

Line 290 appears to be missing a word after "was quoted by." Line 351 has an empty " ," that could be deleted. Line 366 has a "therefore" that should probably be the start of a new sentence.

Author Response

Response to Reviewer 1

This study gives an overview of the production of tobacco and tobacco control development in Mozambique. It effectively combines history, policy, and interviews with key people to fill in the necessary details. I only have a few comments to increase readability and point out potential editing snafus.

(1A) Overall, I don't think that the footnotes serve to make the article easier to understand or read, and the text in them can be either omitted or moved: There is no footnote 1 in the current version, 2 could be moved up to the text of the document, 3 could be an endnote as it is a citation, 5 appears in the text but doesn’t have a footnote at the bottom of the page, 6 could be in the text of the document, 7 could be in the text of the document, 8 could be in the text of the document, and 9 can be deleted because it appears that with current formatting nothing is highlighted in yellow.

-All footnotes were removed and included in the text (3,6, and 8). Edits done on Lines 281, 410, and 446-447 respectively.

-Previous foot note 5 was deleted in text in line 407.

-Previous footnote 9, was deleted and not included in text as the current formatting does not include anything highlighted in yellow. See line 503.

-Previous Footnote 2 was also deleted because of reviewer’s B’s comment (11B) on tobacco exports. See Line 171.

(2A) The second reference to the National Statistics Bureau on line 98 is not capitalized and presents the same shorthand term for the Bureau from line 82. Consider either capitalizing its name throughout or using the shorthand term. Similarly, you gave a shorthand acronym for the Ministry of Agriculture and Rural Development on line 74 but then generally used the full name of the Ministry throughout -- it seems that it might be more efficient to use the shorthand term.

-Change was made in line 164 - The shorthand term for the National Statistics Bureau (INE) is now used throughout.

-Changes were made in lines 180, 334, 359,364 respectively –The shorthand acronym has been used for the Ministry of Agriculture and Rural Development (MADER).

(3A) Table 2 appears to be on the wrong line of the page, it is too far down the page and should be after line 314 and before 315.

To accommodate both Reviewer A and Reviewer B’s comments (comment 28B), table 2 has been excluded and in its place (see line 396), there are just general comments about these companies (lines 386-387).

(4A) Line 290 appears to be missing a word after "was quoted by." Line 351 has an empty " ," that could be deleted. Line 366 has a "therefore" that should probably be the start of a new sentence.

Line 290, which is now line 364 no longer has missing word (replaced by Benfica et al)

The extra comma was deleted on previous line 351, which is now line 425;

Line 366, now line 441 now starts with a new sentence.

Reviewer 2 Report

Review of “An overview of the policy and market landscape of tobacco production and control in Mozambique”

General comments

The paper offers interesting information and data on tobacco in Mozambique but all in all it does not provide a clear picture. Its major deficiency is that no link between tobacco growing and its consumption is discussed and therefore a background of apparent or real conflict between economic and public health interests is not shown. There is a number of questions to be answered such as whether tobacco consumption in Mozambique is on the rise, if tobacco consumption is higher compared to non-tobacco growing countries in Africa, if tobacco farming is related to higher consumption, if farmers smoke more, if they smoke home-processed tobacco, what is prevalence of harm, is it level higher in tobacco growing regions? It would also be of interest to learn if tobacco traders offer discount prices for cigarettes or pay the farmers with processed tobacco. Moreover, no details are given of tobacco control policies. What is taxation policy, what is affordability of tobacco, to what extent a ban on sales for minors is implemented. Are cigarettes available round the clock in grocery shops, are there any vending machines that facilitate purchasing by minors? What control measures are opposed by the economic sector, including national government, local councils, tobacco traders and farmers?

In result, the paper does not answer precisely its initial question where are the tensions and where convergences within the country.

The structure of the paper would be clearer if its analytical part begins with historical section and then followed by general economic context (instead of country context).

In historical section a clear distinction should be made between colonial times, post-independence nationalization of tobacco farming and then its privatization.

Development of trading companies  having concession granted deserves a separate paragraph, including information if these are national or international ones (their country of origin). Potential impact of foreign interests on tobacco policies should also be mentioned.

Detailed comments

47 – All tobacco producing countries like Mozambique have extensive histories of growing that pre-date the rise of a global tobacco control movement.

57 – Consider to add information that supports your claim that Mozambique is “major tobacco growing country”. E.g. Mozambique with its annual production of approx. 100 thousand tons was a tenth (?) producer of tobacco worldwide.

69 – perhaps iv should read: public health involvement (tobacco control)

91- 99 – Quite chaotic paragraph. Some date offer reference to the year while other do not. It should offer data for a fixed year and perhaps some trends over last decades.

102 – Value of tobacco exports in terms of USD is redundant if not related to other similar, comparable data.

117 – Again, the data on two provinces are redundant.

Figure 3 – Perhaps a figure showing proportions at the country level would be more helpful.

132 – When the production reached that volume/

141 – What war? WW Two? Why the post-war and independence periods are combined?

142 – What is expropriated land? Who was deprived of the land?

146 – Was it civil war after independence (after 1974)?

152 – Civil war?

156 – Civil war?

167 – 168 What was rationale behind introducing crops alternative to tobacco?

Table 1 is interesting but redundant.

200 – more likely to have improvement?

226 – What is post-liberalization period in sub-Saharan Africa?

250 – 251 and figure 4 – Please consider to add into class II – producers under contract.

279 – What is the Chifunde concession

287 - 288 – The sentence on tobacco industry contribution should be shifted to the section on general economic context. Moreover, information about its contribution in terms of percentage of revenues would be worthwhile.

294 – 296 In addition to the cost of the processing unit an information should be given about its capacity in terms of a percentage of the total tobacco production.

Table 2 is not informative at all for an external reader. As suggested in more general comments a section on all these companies and their origins would improve the paper.

408 – “The moral principle of public harm” sounds strange. It may be stressed that health has its economic components.

Author Response

Response to Reviewer 2

(1B) The paper offers interesting information and data on tobacco in Mozambique but all in all it does not provide a clear picture. Its major deficiency is that no link between tobacco growing and its consumption is discussed and therefore a background of apparent or real conflict between economic and public health interests is not shown. There is a number of questions to be answered such as whether tobacco consumption in Mozambique is on the rise, if tobacco consumption is higher compared to non-tobacco growing countries in Africa, if tobacco farming is related to higher consumption, if farmers smoke more, if they smoke home-processed tobacco, what is prevalence of harm, is it level higher in tobacco growing regions? It would also be of interest to learn if tobacco traders offer discount prices for cigarettes or pay the farmers with processed tobacco.

The main goal of the paper is to to chart the policy landscape in Mozambique, a major tobacco growing country and a recent Party to the FCTC, using a comprehensive document review process supplemented by analysis of existing economic data such as tobacco production and export. The focus was on charting this landscape by identifying policy and processes in tobacco production and control (See lines 51-63). With our findings we highlight a few of the possible tensions by drawing on existing studies about the intersection between tobacco production and control. The paper was not meant to provide insights into this third link between policy and consumption. We think this is an important area of research and certainly encourage future work in this area in the context of Mozambique. This potential impact is stated on lines 458- 464.

(2B) Moreover, no details are given of tobacco control policies. What is taxation policy, what is affordability of tobacco, to what extent a ban on sales for minors is implemented. Are cigarettes available round the clock in grocery shops, are there any vending machines that facilitate purchasing by minors? What control measures are opposed by the economic sector, including national government, local councils, tobacco traders and farmers?

-Details are on tobacco control decree stated in lines 403-414

-Policies are stated in lines 415-423.

- Further detailed in the supplementary materials, lines 503-504.

 (3B) In result, the paper does not answer precisely its initial question where are the tensions and where convergences within the country.

The objective for the paper has been reworded in lines 61- 63. Conclusion answers question in lines 458-464.

(4B) The structure of the paper would be clearer if its analytical part begins with historical section and then followed by general economic context (instead of country context).

The analytical section now begins with the historical section (line 94) and is followed by general economic/country context (line 156).

(5B) In historical section a clear distinction should be made between colonial times, post-independence nationalization of tobacco farming and then its privatization.

Reworded it as after independence to make the distinction between colonial and post colonial times, please see lines 110 and 121. This will make it easier for the reader to understand.

Please note that Nationalization and privatization of tobacco production all fall under the post independence (or after independence) period.

(6B) Development of trading companies having concession granted deserves a separate paragraph, including information if these are national or international ones (their country of origin). Potential impact of foreign interests on tobacco policies should also be mentioned.
We weren’t entirely clear about what the reviewer was asking in this case. We had noted the entry of China into the tobacco markets of Mozambique and surrounding countries in the paragraph starting on line 396 and in the conclusion section.

Detailed comments

(7B)47 – All tobacco producing countries like Mozambique have extensive histories of growing that pre-date the rise of a global tobacco control movement.

Incorporated the word “all” to sentence, line 47.

(8B) 57 – Consider to add information that supports your claim that Mozambique is “major tobacco growing country”. E.g. Mozambique with its annual production of approx. 100 thousand tons was a tenth (?) producer of tobacco worldwide.

Added: Mozambique ranks fourth as the country that produces the most tobacco in Africa [47]. See line 57-58.

Added new reference 47: FAO Tobacco production quantity (tons): All countries; 2018; See line 618.

(9B)69 – perhaps iv should read: public health involvement (tobacco control)

Now reads public health involvement (tobacco control), see Line 72.

(10B)91- 99 – Quite chaotic paragraph. Some date offer reference to the year while other do not. It should offer data for a fixed year and perhaps some trends over last decades.

Changes made in the following lines respectively: 158, 159, 164,168.

(11B)102 – Value of tobacco exports in terms of USD is redundant if not related to other similar, comparable data

Values have been removed, previous line 102 (Now line 171).

(12B)117 – Again, the data on two provinces are redundant.

Data removed, see line 186.

(13B)Figure 3 – Perhaps a figure showing proportions at the country level would be more helpful.
We have added a new figure (Figure 2) illustrating tobacco production per county/province. Thanks for this suggestion. (see line 188)

(14B)132 – When the production reached that volume

20th century –See lines 96 – 101.

(15B)141 – What war? WW Two? Why the post-war and independence periods are combined?

Removed post-war and just mentioned that it was after independence from the Portuguese Regime to make it clearer to the reader. See lines 110-121. This has been done throughout the text.

(16B)142 – What is expropriated land? Who was deprived of the land?

Now included in the text, the general population was deprived of the land. See lines 111- 112.

(17B)146 – Was it civil war after independence (after 1974)?

Yes, after independence in 1975, the civil war (after independence) lasted precisely from 1977 to 1992. Please see 110 and 121.

(18B)152 – Civil war?

See line 121 –replaced it with after independence.

(19B) 156 – Civil war?

Yes, added “civil war immediately following independence”. Mozambique has had several civil wars. Please see line 125.

(20B)167 – 168 What was rationale behind introducing crops alternative to tobacco?

The rationale has now been included in the introduction in lines 61-63.

(21B) Table 1 is interesting but redundant.
We have decided to retain Table 1 as we think it provides important comparative information on crops produced in Mozambique. We have added further rationale for this table in the text on lines 62 and 144.

(22B)200 – more likely to have improvement?

Restructured to “show improvement”, now line 272.

(23B)226 – What is post-liberalization period in sub-Saharan Africa?

General economic post liberalization, because the era is vast, “post –liberation period” has been removed. Please see 298-299.

(24B) 250 – 251 and figure 4 – Please consider to add into class II – producers under contract.

Producers under contract have been added onto class II on figure 5. See Line 339.

(25B)279 – What is the Chifunde concession

A concession in Chifunde district in Tete province. As stated in line 337, Chifunde was the largest concession controlled by Alliance One, and its loss jeopardized the viability of its work in Mozambique. 

(26B) 287 - 288 – The sentence on tobacco industry contribution should be shifted to the section on general economic context. Moreover, information about its contribution in terms of percentage of revenues would be worthwhile.

The sentence has been shifted to the general economic context session in lines 169-170.

(27B)294 – 296 In addition to the cost of the processing unit an information should be given about its capacity in terms of a percentage of the total tobacco production.

No data on yearly production.

(28B) Table 2 is not informative at all for an external reader. As suggested in more general comments a section on all these companies and their origins would improve the paper.

 Table 2 has been excluded and more general comments were included as suggested in lines 386-387.

(29B)408 – “The moral principle of public harm” sounds strange. It may be stressed that health has its economic components.

 Removed “moral principle of public harm” and just left health protection considerations. See line 490 in track change.

Round 2

Reviewer 2 Report

I read the manuscript again and responses to my comments but it required extra efforts on my side as lines referred to in the responses did not correspond with the lines on the manuscript I received.

My general impression is that the paper has improved but still some points need further clarifications.  Some paragraphs in its new historical part are still chaotic and do not offer a clear picture of historical developments. As an example I would quote  following sentences: ”During the independence period from the Portuguese regime after 1975, tobacco farms flourished in central Mozambique largely because farmers used inexpensive forced labour and expropriated land from locals. Although tobacco leaf is currently one of  main agricultural exports, its contribution to exports never exceeded 2% between 1955-1964”. Both sentences are not logically associated. Moreover, the first one refers to post-colonial period while the second one to colonial times that are compared with current contribution of tobacco to the national exports. The comparison seems to be deficient as there are no data what is the current contribution of tobacco in the national exports but only its share in exports of agricultural products

The sentences that follow in historical section need to offer more precise distinction of the different stages of tobacco farming after independence. When tobacco farming  flourished? when state farms were established, when shut down, when a system of concessions was re-introduced.

Toward the end of that section the changes in crops structure in Zambezi province are presented while a table that follows shows changes at the national level which are not disused at all.

The section on the country context improved but still is not satisfactory. E.g. information that agriculture sector share in GDP has reduced is presented as a symptom of economic crisis while it could be seen as positive economic development. Instead of presenting the tobacco industry contribution n monetary terms its percentage share in national tax revenues should be presented.

It could found at the end of the paper what acronym MADER is for but its full name has to be given before using its acronym form.

My crucial question was if MLT was purely national company or was associated with international tobacco producers which helped MLT to achieve almost monopolistic position as well as in its corporate social responsibility activities. In general, a process of formation of the concession system needs more detailed description, how many companies acted in the beginning, how the process of concentration has progressed and in which years.

Author Response

Thank you for your generous feedback. We have address each point in the attached word document.
